# Diagnostic Efficacy of C-Reactive Protein in Differentiating Various Causes of Exudative Pleural Effusion: Disease Research Should Not Be Exclusive to the Wealthy [note 1]

**DOI:** 10.3390/arm93040029

**Published:** 2025-08-05

**Authors:** Majed Odeh, Yana Kogan, Edmond Sabo

**Affiliations:** 1Department of Internal Medicine A, Bnai Zion Medical Center, Haifa 3104802, Israel; janakgan37@gmail.com; 2Faculty of Medicine, Technion—Israel Institute of Technology, Haifa 3109601, Israel; edmondsa@clalit.org.il; 3Pulmonary Division, Carmel Medical Center, Haifa 3436212, Israel; 4Institute of Pathology, Carmel Medical Center, Haifa 3436212, Israel

**Keywords:** C-reactive protein, pleural effusion, exudates, parapneumonic effusion, malignant effusion, tuberculous effusion

## Abstract

**Highlights:**

**What are the main findings?**
Serum and pleural fluid C-reactive protein (CRPs, CRPpf) and their gradient (CRPg) demonstrated significant diagnostic value, while their ratio (CRPr) did not, in discrimination between various causes of exudative pleural effusion (PE).

**What is the implication of the main finding?**
Despite initial evaluations including thoracentesis, pleural fluid analysis, and closed pleural biopsy, 20–40% of exudative PEs are still undiagnosed. Furthermore, many patients are unwilling or unsuitable to undergo thoracoscopy and open pleural biopsy for their invasive nature and risk of complications, and these procedures are not widely available for most of the developing world’s population. In these situations, CRPs, CRPpf, and CRPg can be helpful in managing these patients.Even in patients who cannot undergo needle pleural paracentesis, a simple and inexpensive test, such as measuring CRPs level, can be helpful in evaluating the etiology of their PE, and in directing their management.

**Abstract:**

**Background and Objectives**: Discrimination between various causes of exudative pleural effusion (PE) remains a major clinical challenge, and to date, definitive biochemical markers for this discrimination remain lacking. An increasing number of studies have reported that serum C-reactive protein (CRPs), pleural fluid CRP (CRPpf), and CRPpf/CRPs ratio (CRPr) are useful for the differential diagnosis of exudative PE; however, their efficacy rate is not similar in these studies. The majority of these studies were conducted on small groups of subjects, and the efficacy of the gradient between CRPs and CRPpf (CRPg—calculated as CRPs—CRPpf) in this differentiation has not been previously investigated. This study aims to evaluate the efficacy rate of CRPs, CRPpf, CRPg, and CRPr in the differential diagnoses of various causes of exudative PE in a relatively large cohort of patients. **Materials and Methods**: The research group included 282 subjects with exudative PE—146 had parapneumonic effusion (PPE), 126 had malignant pleural effusion (MPE), and 10 had tuberculous pleural effusion (TPE). The values are presented as mean ± SD. **Results**: The mean CRPs level was significantly higher in the PPE group compared to the MPE group (*p* < 0.0001) and the TPE group (*p* < 0.001), and also significantly higher in the TPE group than in the MPE group (*p* = 0.0009). Similarly, the mean CRPpf level was significantly higher in the PPE group than in the MPE group (*p* < 0.0001) and the TPE group (*p* = 0.04), and also significantly higher in the TPE group than in the MPE group (*p* < 0.0001). The mean CRPg level was significantly higher in the PPE group than in both the MPE group (*p* < 0.0001) and the TPE group (*p* < 0.002). The mean CRPr level did not differ significantly among these groups of exudate. **Conclusions**: CRPs, CRPpf, and CRPg are effective in the differential diagnosis of exudative PE, while CRPr was not effective in this regard. The main limitation of this study is that the sample size of the TPE group is very small.

## 1. Introduction

Pleural effusion (PE) is a common clinical condition, with a global prevalence of approximately 400 cases per 100,000 people. In the United States alone, nearly 1.5 million individuals develop PE each year. The principal and first step in the evaluation of PE is to characterize it as a transudative effusion or an exudative effusion. Currently, the gold standard method in this characterization is based on Light’s criteria [1]. However, these criteria are not helpful in determining the underlying cause of an exudative PE, which is significantly more challenging to diagnose than transudative PE. To date, definitive biochemical markers for distinguishing between the various causes of exudative PE are lacking. This diagnostic challenge imposes a significant burden on both the healthcare system and the patients [2,3,4,5].

Pneumonia, cancer, and tuberculosis are the causes of most pleural exudates. An exudative PE usually necessitates a much more comprehensive and invasive diagnostic procedure than transudative PE. After initial evaluation with thoracentesis, extensive pleural fluid analysis and closed pleural biopsy, about 20–40% of exudative PEs remain undiagnosed [6,7,8,9,10,11,12]. These effusions are concerning because, upon follow-up with more invasive procedures, many are found to be malignant or tuberculous, depending on whether the region is endemic for tuberculosis or not [6,12,13,14,15,16,17,18,19]. Indeed, cytological examinations of the pleural fluid may result in missed or incorrect diagnoses, are associated with remarkably high false-negative rates (in about 40% of cases), and are time-consuming. The diagnosis of tuberculous PE (TPE) by simple thoracentesis analysis is usually difficult. The pleural fluid GeneXpert test, culture, and Ziehl–Neelsen staining for *Mycobacterium tuberculosis* have limited diagnostic utility, as GeneXpert shows low sensitivity and both staining and culture are time-consuming and frequently yield negative results [2,3,4,5,20,21]. Although pleural fluid adenosine deaminase (ADA) with a cut-off level of ≥35–40 U/L can be highly sensitive for diagnosing TPE, its specificity is limited, as elevated ADA levels are also common in PPE, particularly in complicated cases, and in MPE, especially lymphomatous types [22,23,24,25]. The diagnostic yield of closed pleural biopsy is not sufficiently high, and its complications are not uncommon [19,26,27,28,29,30,31,32]. Positive cultures from pleural fluid give definitive evidence of parapneumonic effusion (PPE), but they are positive in about 60% of cases, and they do not give rapid results. Pleural fluid pH level is typically lower in PPE, particularly in complicated cases; however, low pH levels may also be observed in TPE and MPE. An increased pleural fluid leukocyte count with neutrophilic predominance is typical of PPE; however, neutrophilic predominance may also be observed, though less commonly, in MPE and in the acute phase of TPE. Moreover, pleural fluid glucose levels below 60 mg/dL may be observed in PPE, TPE, and, though less commonly, in MPE [2,3,4,5,20,21]. Many other markers, such as tumor markers and cytokines, have been proposed for distinguishing between various causes of exudative PE; however, their sensitivity and specificity are not sufficiently high to enable a definitive diagnosis [2,3,4,5,20,21].

Invasive procedures like thoracoscopy and open pleural biopsy are considered the golden standard for diagnosing the etiology of exudative PE, and yield excellent results, particularly in cases of MPE and TPE. Yet, many patients are unwilling or unsuitable to undergo these procedures due to their invasive nature and risk of complications, and additionally, these procedures are expensive and time-consuming [2,3,4,5,20,21]. Furthermore, these procedures are not widely available for most of the developing world’s population due to lack of resources and expertise, and even despite these diagnostic invasive procedures, about 10–30% of exudative PEs remain undiagnosed [6,9,33,34,35,36,37,38,39].

A hindrance in the diagnosis of exudative PE and initiation of appropriate treatment usually leads to increased morbidity and mortality rates. Therefore, highly accurate, and non-invasive diagnostic tools, which are not expensive, simply measured, and have high specificity and sensitivity rates, are needed in the differential diagnosis of exudative PE, particularly in patients who are unwilling or unsuitable to undergo invasive procedures, or in so many patients worldwide where these procedures are not available for them.

An increasing number of studies have reported that serum C-reactive protein (CRPs), pleural fluid CRP (CRPpf), and CRPpf/CRPs ratio (CRPr) are helpful in the diagnosing the etiology of exudative PE [40,41,42,43,44,45,46,47,48,49,50,51,52,53,54,55,56,57,58,59,60,61,62,63,64,65,66,67,68,69,70,71,72]. The majority of these studies were conducted on small sample sizes; some of them measured and analyzed both CRPpf and CRPs [40,41,42,43,44,45,46,47,48,49,50,51,52], some only CRPpf [53,54,55,56,57,58,59,60,61,62,63], and some only CRPs [20,64,65,66,67,68,69,70,71,72] for this differential diagnosis. The diagnostic strength of these three markers is not the same among these studies, and some of them have unreasonable results where they demonstrated levels of CRPr > 1 [41,44,45,51], which is impossible. To date, the diagnostic role of the CRPs to CRPpf gradient (CRPg) has been assessed only in differentiating transudative from exudative PE [73], and in differentiating uncomplicated from complicated PPE [74], where it has proven to be highly useful. In the differential diagnosis of exudative PE, this parameter was not previously reported.

The aim of this retrospective study, conducted on a relatively large cohort of 282 patients with exudative PE—a sample size larger than that of most previous studies—is to evaluate the diagnostic efficacy of serum CRPs, RPpf, CRPg, and CRPr in differentiating between various causes of exudative PE.

## 2. Patients and Methods

### 2.1. Patients

The study population consisted of 282 patients with exudative PE who were admitted to the Department of Internal Medicine at Bnai Zion Medical Center between January 2000 and October 2016. All patients underwent pleural fluid drainage and comprehensive diagnostic evaluation to determine the underlying etiology of their PE, resulting in a definitive diagnosis. Of these, 146 patients (aged 24–92 years) were diagnosed with PPE, 126 (aged 29–95 years) with MPE, and 10 (aged 23–86 years) with TPE. The diagnoses of PPE, MPE, and TPE were established based on internationally accepted criteria, as described in our recently published study [73]. Any subject with exudative PE who had any comorbidity which may significantly influence CRPs and CRPpf was not included in our study.

### 2.2. Methods

Relevant data were extracted from the patients’ medical records. Subjects included in this study were those with a definitive diagnosis of PE as PPE, MPE, or TPE, and who had undergone measurement of both CRPpf and CRPs levels. CRP levels were measured using a Cobas c 501 analyzer (Roche Diagnostics), following the manufacturer’s instructions. The research was performed in agreement with the Declaration of Helsinki, and the research protocol was approved by the Bnai Zion Medical Center Ethics Committee (0107-16-BNZ, 25 October 2016). CRPg was calculated as CRPs–CRPpf, and CRPr was calculated as CRPpf/CRPs.

### 2.3. Statistical Analysis

Data were analyzed using SPSS software, ver. 23 (IBM Corporation, Armonk, NY, USA). Values of various parameters are presented as means ± standard deviation (SD) of means with their 95% confidence intervals (CIs). The Kolmogorov–Smirnov test was used to assess the normality of data distribution. Unpaired Student’s *t*-tests were applied to compare parametric groups. Bonferroni correction was used to adjust *p*-values for multiple comparisons. Receiver operating characteristic (ROC) curve analysis was performed to determine optimal cut-off values for distinguishing between groups. These cut-off values, selected based on the highest total accuracy, were established using the Youden index, which balances sensitivity and specificity. Sensitivity, specificity, negative predictive value (NPV), positive predictive value (PPV), total accuracy, area under the ROC curves (AUC), and odds ratios were calculated. The significance of the best cut-off values was achieved as needed by Fisher’s exact test or the χ^2^ test. We considered *p*-values statistically significant if they were ≤0.05.

## 3. Results

The PPE group included 146 subjects, the MPE group included 126 subjects, and the TPE group included 10 subjects. In all cases, pleural fluid CRP (CRPpf) levels were lower than serum CRP (CRPs) levels. The mean age of the MPE group was significantly higher than that of the PPE group (75.9 ± 10.2 years vs. 69.6 ± 16.7 years; *p* < 0.003) and the TPE group (66.1 ± 19.2 years; *p* < 0.0009). No significant age difference was observed between the PPE and TPE groups (*p* = 0.53) (Table 1).

CRPs, CRPpf, CRPg, and CRPr mean levels are shown in Table 1 and Figure 1, Figure 2, Figure 3 and Figure 4. The CRPs mean level was higher significantly in the PPE group, 215.0 ± 102.9 mg/L (95% CI: 198.5–232.2), than in the MPE group, 56.1 ± 39.5 mg/L (95% CI: 23.3–71.0) (*p* < 0.0001), and the TPE group, 98.7 ± 12.9 mg/L (95% CI: 91.4–107.9) (*p* < 0.003), and in the TPE group than in the MPE group (*p* = 0.0009) (Table 1, Figure 1).

The CRPpf mean level was higher significantly in the PPE group [82.7 ± 58.5 mg/L (95% CI: 73.1–92.2)] than in the MPE group [18.9 ± 13.9 mg/L (95% CI: 7.6–28.1)] (*p* < 0.0001), and the TPE group [45.0 ± 9.4 mg/L (95% CI: 38.3–51.7)] (*p* = 0.04), and in the TPE group than in the MPE group (*p* < 0.0009) (Table 1, Figure 2).

The CRPg mean level was higher significantly in the PPE group [132.7 ± 71.2 mg/L (95% CI: 122.1–148.4)] than in the MPE group [37.2 ± 29.2 mg/L (95% CI: 13.2–45.4)] (*p* < 0.0001), and in the TPE group [53.7 ± 18.6 mg/L (95% CI: 40.4 ± 67.0)] (*p* < 0.002). No significant difference was found between the TPE group and the MPE group (*p* = 0.08) (Table 1, Figure 3).

Regarding CRPr mean level, a significant difference was not found between the three groups (*p* > 0.05) (Table 1, Figure 4).

The best CRPs, CRPpf, and CRPg cut-off values (calculated by the ROC analysis) were used for differentiation between the three groups, and their relevant statistical parameters are shown in Figure 1, Figure 2, Figure 3, Figure 4, Figure 5, Figure 6, Figure 7, Figure 8 and Figure 9. The best CRPs cut-off value for the differentiation between PPE and MPE was 107.5 mg/L, with a sensitivity of 81%, specificity of 87.3%, AUC of 93.7%, and *p* < 0.0001 (Figure 1).

The best CRPs cut-off value for the differentiation between PPE and TPE was 119 mg/L, with a sensitivity of 75%, specificity of 100%, AUC of 83%, and *p* < 0.0001 (Figure 2).

The best CRPs cut-off value for the differentiation between MPE and TPE was 79.5 mg/L, with a sensitivity of 100%, specificity of 74%, AUC of 82.4%, and *p* < 0.0001 (Figure 3).

The best CRPpf cut-off value for the differentiation between PPE and MPE was 33.5 mg/L, with a sensitivity of 83%, specificity of 84%, AUC of 92.2%, and *p* < 0.0001 (Figure 4).

The best CRPpf cut-off value for the differentiation between PPE and TPE was 56.5 mg/L, with a sensitivity of 52%, specificity of 100%, AUC of 63.5%, and *p* = 0.001 (Figure 5).

The best CRPpf cut-off value for the differentiation between MPE and TPE was 28.5 mg/L, with a sensitivity of 100%, specificity of 79%, AUC of 92.9%, and *p* < 0.0001 (Figure 6).

The best CRPg cut-off value for the differentiation between PPE and MPE was 72.5 mg/L, with a sensitivity of 76%, specificity of 85%, AUC of 91.7%, and *p* < 0.0001 (Figure 7).

The best CRPg cut-off value for the differentiation between PPE and TPE was 53.5 mg/L, with a sensitivity of 90%, specificity of 70%, AUC of 87.4%, and *p* < 0.0001 (Figure 8).

The best CRPg cut-off value for the differentiation between MPE and TPE was 32.5 mg/L, with a sensitivity of 100%, specificity of 61%, AUC of 72.4%, and *p* < 0.0001 (Figure 9).

## 4. Discussion

CRP is one of the main acute-phase reactants, produced mainly in the liver by the hepatocytes, and in excess, during inflammation and tissue injury. Significant tissue injury and various inflammatory processes of either noninfectious or infectious etiology increase its level in the blood [75,76]. Several mediators of inflammation, particularly interleukin-6 and tumor necrosis factor-α, trigger induction of its production by the hepatocytes. The level of this main acute-phase reactant in pleural fluid depends mainly on its blood level. Therefore, the CRPpf level predominantly reflects the CRPs level, but it never equals or exceeds it. This fact was demonstrated in all the patients in this study and in most of the previously reported studies [40,41,42,43,44,45,46,47,48,49,50,51,52]. An increasing number of studies in recent years have investigated the efficacy of CRPs and CRPpf in differentiating between different exudative PEs. Many of them have demonstrated that both CRPs and CRPpf are helpful in this differentiation, but their efficacy rate was not similar in these studies [40,41,42,43,44,45,46,47,48,49,50,51,52,53,54,55,56,57,58,59,60,61,62,63,64,65,66,67,68,69,70,71,72].

Results of the present study showed that CRPs and CRPpf are helpful in differentiating between the main three causes of exudative PE. CRPs was somewhat stronger than CRPpf in this differentiation. The CRPs mean level was significantly higher in the PPE group than in the MPE group (*p* < 0.0001) and the TPE group (*p* < 0.003), and in the TPE group than in the MPE group (*p* = 0.0009) (Table 1, Figure 1). With the best CRPs cut-off value of 107.5 mg/L, a strong differentiation between PPE and TPE was achieved: the AUC was 93.7%, the sensitivity was 81%, and the specificity was 87.3%, with *p* < 0.0001 (Figure 1). With the best CRPs cut-off value of 119 mg/L, a strong differentiation between PPE and TPE was achieved: the AUC was 83%, the sensitivity was 75%, and the specificity was 100%, with *p* < 0.0001 (Figure 2). With the best CRPs cut-off value of 79.5 mg/L, a strong differentiation was also achieved between MPE and TPE: the AUC was 82.4%, the sensitivity was 100%, and the specificity was 74%, with *p* < 0.0001 (Figure 3). These results are similar to results of most other relevant studies [6,8,9,10,11,12,13,14,15,16,17,18,19,20,32,33,34,35,36,37,38,39], but are stronger and more accurate than most of them regarding the efficacy rate of CRPs in differentiating between these three groups of exudative PE.

The CRPpf mean level was significantly higher in the PPE group than in the MPE group (*p* < 0.0001) and the TPE group (*p* = 0.04), and in the TPE group than in the MPE group (*p* < 0.0009) (Table 1, Figure 2). With the best CRPpf cut-off value of 33.5 mg/L, a strong differentiation between PPE and MPE was achieved: the AUC was 92.2%, the sensitivity was 83%, and the specificity was 84%, with *p* < 0.0001 (Figure 4). With the best CRPpf cut-off value of 56.5 mg/L, a good differentiation between PPE and TPE was achieved: the AUC was 63.5%, the sensitivity was 52%, and the specificity was 100%, with *p* = 0.001 (Figure 5). With the best CRPpf cut-off value of 28.5 mg/L, a strong differentiation was achieved between MPE and TPE: the AUC was 92.9%, the sensitivity was 100%, and the specificity was 79%, with *p* < 0.0001 (Figure 6). These results are similar to results of most other relevant studies [8,9,10,11,12,13,14,15,16,17,18,19,20,21,22,23,24,25,26,27,28,29,30,31], but are stronger and more accurate than most of them regarding the efficacy rate of CRPpf in differentiating between these three groups of exudative PE.

This study is the first to investigate the efficacy rate of CRPg in differentiating between various causes of exudative PE. This marker has previously been evaluated in only two contexts: distinguishing between transudative and exudative PE [73], and between complicated and uncomplicated PPE [74]. In both cases, it demonstrated high diagnostic utility. Our results showed that CRPg is also helpful for differentiating between various causes of exudative PE. The CRPg mean level was significantly higher in the PPE group than in the MPE group (*p* < 0.0001) and in the TPE group (*p* < 0.002). No significant difference was found between the MPE group and the TPE group (*p* = 0.08) (Table 1, Figure 3). With the best CRPg cut-off value of 72.5 mg/L, a strong differentiation was achieved between PPE and MPE: the AUC was 91.7%, the sensitivity was 76%, and the specificity was 85%, with *p* < 0.0001 (Figure 7). With the best CRPg cut-off value of 53.5 mg/L, a strong differentiation was also achieved between PPE and TPE: the AUC was 87.4%, the sensitivity was 90%, and the specificity was70%, with *p* < 0.0001 (Figure 8). Since there was a trend for this marker to differentiate MPE from TPE (*p* = 0.08), we performed the ROC analysis for this differentiation. With the best CRPg cut-off value of 32.5 mg/L, a good differentiation between MPE and TPE was achieved: the AUC was 72.4%, the sensitivity was 100%, and the specificity was 61%, with *p* < 0.0001 (Figure 9). According to these results, we believe that the absence of statistical significance between mean value of CRPg of the MPE group and that of the TPE group is, most probably, related to the very small cohort of the TPE group. Our findings support the need for further prospective and retrospective studies with larger patient cohorts to investigate the role of this marker in differentiating between various causes of exudative PE.

Regarding mean level of CRPr, a significant difference was not found between the three groups of exudative PE (Table 1, Figure 4), indicating that CRPr is not helpful in differentiating between these three groups, as has been recently demonstrated in differentiating between transudative and exudative PE [73], and between complicated and uncomplicated PPE [74]. There are eight published previous studies, which investigated the efficacy of CRPr in differentiating between various groups of exudative PE [40,41,42,43,44,45,47,51]. Our results are similar to results of three of them [40,43,47] and oppose results of only one study [10], which found a significant difference between MPE group and TPE group regarding CRPr. The other four studies [41,44,45,51], which all had small cohorts of patients, demonstrated unreasonable results because they achieved levels of CRPr > 1, which is impossible. The CRPpf level is consistently lower than the CRPs in each patient, as was demonstrated in so many, hundreds if not thousands, other studies, including ours. As stated above, similar to albumin, CRP is synthesized primarily by hepatocytes, and its pleural fluid level depends largely on its serum concentration. It can also be produced, but in very small amounts, in various extrahepatic sites, including the pleura and pleural cavity, by various cells, including some inflammatory cells [75,76]. Its local production in the pleura and pleural cavity is so minimal that it cannot increase the CRPpf level to equal or exceed the CRPs level [75,76]. Therefore, CRPpf levels are always lower than those of CRPs, exactly as with the case of albumin, where hundreds, if not thousands, of studies have demonstrated the same findings regarding its serum and pleural fluid levels. Anyway, the role of CRPr in differentiating between various causes of exudative PE should be evaluated in additional large cohorts of studies.

Although the mean age of the MPE group was significantly higher than that of the PPE group and the TPE group, and although CRPs may increase with age by about 1 mg/L, this increase is very small and the CRPs level remains within its normal range, which is < 5 mg/L. Thus, age does not significantly influence levels of CRPs [77].

The optimal approach in the evaluation of a patient with exudative PE is to achieve definitive diagnosis of its etiology rapidly in order to direct the patient’s management accordingly and properly. However, for many patients worldwide, particularly in most of the developing countries, this optimal approach is significantly limited because the gold standard expensive diagnostic procedures are not available for most of them due to lack of resources and expertise. Even in developed countries, many patients are unwilling or unsuitable to undergo these procedures. In this situation, many of these patients, particularly in endemic regions of tuberculosis, are treated empirically by depending on the clinical picture, available imaging modalities, and on relevant biochemical markers without achieving a definitive diagnosis of their PE etiology [14,17,19,78,79,80,81,82]. Pleural fluid ADA is considered the most useful biomarker for TPE diagnosis because of its high sensitivity rate with a level of ≥35–40 U/L, and in endemic regions of tuberculosis, ADA is crucial in making decisions regarding starting empirical treatment for tuberculosis without a definitive diagnosis [14,17,19,78,79,80,81,82]. A major limitation of pleural fluid ADA is its low specificity, as levels ≥ 35–40 U/L are also commonly observed in PPE, particularly in complicated cases, and in MPE, especially those of lymphomatous origin [22,23,24,25,78,81,83]. Indeed, in a large series of 2014 patients with PE from Spain, Porcel et al. reported pleural fluid ADA level > 35 U/L in 70% of empyemas, in 44% of non-purulent complicated PPE, and in 57% of lymphomatous PE. Furthermore, they demonstrated that very high levels of pleural fluid ADA > 250 U/L are highly suggestive of empyema and lymphoma rather than tuberculosis [83]. If this is the situation, CRP could be helpful and a strong biomarker for the differential diagnosis of exudative PE. Unfortunately, our TPE group was very small, but our encouraging results of this study, and the data presented above in this paragraph, strongly stimulate performing further studies on CRP with large cohorts of exudative PE, particularly those with TPE.

Limitations of this study are as follows: (1) It is a retrospective study. (2) The cohort of the TPE group is very small, but this is because tuberculosis and tuberculous pleurisy are uncommon in our country. (3) Other potential causes of exudative PE, such as autoimmune diseases, are lacking. This is because PE due to autoimmune diseases, which necessitates drainage, is rare [84,85], and we did not find patients with this entity in our study.

## 5. Conclusions

The results of the present study demonstrated the high diagnostic value for CRPs and CRPpf in differentiating between various causes of exudative PE, whereas the CRPr was not useful for this purpose. These results are very important and could be helpful for patients with exudative PE, particularly those who are unwilling or unsuitable to undergo invasive procedures, and those who do not have access to these expensive procedures, who may represent the majority of the developing world’s population. Even in patients who cannot undergo pleural paracentesis, simple and inexpensive tests, such as measuring the CRPs level, can be helpful in evaluating the etiology of their exudative PE, and can strongly direct the management of these patients. The present study also showed, for the first time, a high efficacy rate of CRPg in this differentiation, at least in differentiating between PPE and MPE, and between PPE and TPE. Our findings stimulate performing further studies on big groups of subjects for establishing the high efficacy rate of CRPs, CRPpf, and CRPg in differentiating between various causes of exudative PE, and for investigating CRPr validity in this differentiation. Finally, it should be emphasized that disease research should not be exclusive to the wealthy, and it should be directed in order to also be practical for populations in low-resource settings who do not have access to expensive and invasive diagnostic procedures, and who represent a substantial portion of the world’s population. Furthermore, it should also be emphasized that diagnostic evaluation guidelines should be arranged in an adequate manner, making these guidelines also practical for these populations.

## Data Availability

Data are available from the corresponding author upon reasonable request.

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
