# Peer review of "Diagnostic Efficacy of C-Reactive Protein in Differentiating Various Causes of Exudative Pleural Effusion: Disease Research Should Not Be Exclusive to the Wealthy†"

_arm, 2025, doi:10.3390/arm93040029_

Round 1

Reviewer 1 Report

Comments and Suggestions for Authors

- The main title is too long and merges two different ideas: the scientific part “Diagnostic Efficacy…” and a sociopolitical phrase “Disease Research Should Not Be Only for Rich.” This affects precision and scholarly tone. Perhaps both ideas should be reframed or the second part presented as a subtitle or explained in the introduction.

- The statement “These authors contributed equally to the work” is accurately conveying the idea, but can be executed in a more straightforward manner. We suggest placing † right after the authors' names and connecting it to a footnote.

- The “Highlights” section does not demonstrate command of grammar. As an illustration, “their ration has no role” is incorrectly stated instead of “their ratio has no role.” Mistakes in spelling in scientific writing are damaging to credibility.

– In the implications, the wording within the sentences is complex and intricate. For example, “Since after initial evaluation of exudative PE…” is quite off. An improvement could be: “Despite initial evaluations including thoracentesis, pleural fluid evaluation, and closed pleural biopsy, 20-40%of exudative PEs are still undiagnosed..."

- The last bullet point ‘Highlights’ should be more succinct and detailed in two sentences. In addition, “simple and inexpensive test, as measuring CRPs level,” changes to “a simple and inexpensive test, such as measuring CRP levels…”.

- Including the same information within the abstract and highlights is inefficient. Condensing or providing references in summary format can be more helpful.

- Abstracts contain the term ‘researches’. It should be edited to ‘studies’ in the entire document.

- Replace the phrase ‘until yet’ to ‘to date’ or ‘so far’ as it does not follow the English standards.

- Providing too much numerical data in the abstract makes comprehension difficult. Abstracts should focus more on providing significant statistics.

- There are typographical mistakes, “ME group” should change to “MPE group” in line 62.

- The nonstandard conclusion phrase: “are much efficacious” should be changed to “are effective” or “demonstrated significant diagnostic value.”

- The abstract includes no statement regarding the limitations of the study. Acknowledging at least something, such as the TPE sample size, would enhance clarity.

Reviewer 2 Report

Comments and Suggestions for Authors

The article is very interesting, but I have some technical and substantive comments.

  1. Group of TPE contained 10 subjects while the other two groups contain significantly more data.So can the results be considered sufficiently representative?
  2. The authors use a very strange convention in presenting p-values.They use a lot of specific symbols, which are then described in captions under tables and figures.It is very illegible and makes reading difficult.I suggest rebuilding Table 1. It is also worth presenting the results in the form of boxplots, but with p-values ​​clearly marked.Please, for example, read this study (or any other, there are many of them on the Internet)https://www.sthda.com/english/articles/24-ggpubr-publication-ready-plots/76-add-p-values-and-significance-levels-to-ggplots/
  3. Doubts are raised by ROC curves, which were probably not drawn correctly. The ROC curve ALWAYS starts in the lower left corner and ends in the upper right corner. In Figures 2, 3, 5, 9 this rule is probably not followed. In addition, I suggest that the cut-off values ​​be plotted directly on the ROC curve and not just given in the caption under the figure.
  4. In addition, please clarify what the PPV and NPV measures are.
  5. In point 2.3 I read "The significance of the best cut-off values ​​was achieved as needed, by Fisher's exact test or the χ2 test." I can't find even the slightest mention of this in the text. Please complete it as a must.
  6. In Figures 1-4 I suggest introducing colors and changing them to classic boxplots.

Reviewer 3 Report

Comments and Suggestions for Authors

Diagnostic Efficacy of C-Reactive Protein in Discrimination between Various Causes of Exudative Pleural Effusion

This retrospective study explores the diagnostic utility of serum and pleural fluid C-reactive protein levels, and their derived gradients and ratios, in distinguishing causes of exudative pleural effusion. The findings contribute meaningfully to resource-limited diagnostic strategies. However, several areas require clarification or major revisions before publication to enhance the manuscript's impact and reproducibility.

Here are some suggestions to improve the manuscript:

Line #

Comments

Title

Diagnostic Efficacy of C-Reactive Protein in Discrimination between Various Causes of Exudative Pleural Effusion: Disease Research Should not be Only for Rich

OR

Diagnostic Efficacy of C-Reactive Protein in Differentiating Various Causes of Exudative Pleural Effusion: Disease Research Should Not Be Exclusive to the Wealthy

For better clarity and readability

Abstract

43

…definitive biochemical markers for this discrimination are lacking.

OR                                        

…definitive biochemical markers remain lacking.

46

…exudative PE, however their efficacy rate…

OR

…exudative PE, however, their efficacy rate…

Add comma

47-49

The majority of these studies was performed on small groups of subjects, and gradient of CRPs and CRPpf (CRPg - calculated as CRPs - CRPpf) efficacy in this differentiation was not investigated before.

OR

The majority of these studies were conducted on small groups of subjects, and the efficacy of the gradient between CRPs and CRPpf (CRPg, calculated as CRPs − CRPpf) has not been previously investigated.

49

Our aim in this research is evaluation of the efficacy rate…

OR

This study aims to evaluate the efficacy rate…

52

…with exudative PE – 146…

OR

…with exudative PE: 146…

Punctuation

55-61

CRPs mean level was significantly higher in the PPE group than in the MPE group (p < 0.0001) and in the TPE group (p < 0.001); and in the TPE group than in the MPE group (p = 0.0009). CRPpf mean level was significantly higher in the PPE group than in the MPE group (p < 0.0001) and in the TPE group (p = 0.04); and in the TPE group than in the MPE group (p < 0.0001). CRPg mean level was significantly higher in the PPE group than in the MPE group (p < 0.0001) and in the TPE group (p < 0.002).

OR

The mean CRPs level was significantly higher in the PPE group compared to the MPE group (p < 0.0001) and the TPE group (p < 0.001), and also significantly higher in the TPE group than in the MPE group (p = 0.0009). Similarly, the mean CRPpf level was significantly higher in the PPE group than in the MPE group (p < 0.0001) and the TPE group (p = 0.04), and also significantly higher in the TPE group than in the MPE group (p < 0.0001). The mean CRPg level was significantly higher in the PPE group than in both the MPE group (p < 0.0001) and the TPE group (p < 0.002).

Sentence structures can be improved for clarity, consistency and enhanced readability.

61

CRPr mean level did not differ significantly…

OR

The mean CRPr level did not differ significantly…

62

CRPs, CRPpf, and CRPg are very effective…

OR

CRPs, CRPpf, and CRPg are effective…

63

…CRPr is not effective…

OR

…CRPr was not effective…

1.      Introduction

80-82

Pleural effusion (PE) is a common complication in clinical field, has a prevalence of about 400 per 100,000 people worldwide, and each year in the United States alone; almost 1.5 million inhabitants evolve PE.

OR

Pleural effusion (PE) is a common clinical condition, with a global prevalence of approximately 400 cases per 100,000 people. In the United States alone, nearly 1.5 million individuals develop PE each year.

85-86

However, these criteria are not helpful in diagnosing the cause of an exudative PE, which is much more difficult than for transudative PE.

OR

However, these criteria are not helpful in determining the underlying cause of an exudative PE, which is significantly more challenging to diagnose than transudative PE.

86-90

To date, definitive biochemical markers for differentiation between various causes of exudative PE are lacking, and this differentiation represents a major medical challenge associated with big expenditures to the health care system and to the patients.

OR

To date, definitive biochemical markers for distinguishing between the various causes of exudative PE are lacking. This diagnostic challenge imposes a significant burden on both the healthcare system and patients.

94

…extensive analysis of the pleural fluid...

OR

…extensive pleural fluid analysis…

80

...can be transmitted from poultry to human beings via poultry origin food-products...

OR

...can be transmitted from poultry to humans through poultry-origin food products...

95-97

These effusions are worrying, because on follow-up using more invasive procedures, most of them will be malignant or tuberculous depending on the specific region if it is endemic for tuberculosis or not.

OR

These effusions are concerning because, upon follow-up with more invasive procedures, many are found to be malignant or tuberculous, depending on whether the region is endemic for tuberculosis or not.

99

…pleural fluid may lead to missed diagnosis or misdiagnosis…

OR

…pleural fluid may result in missed or incorrect diagnoses…

103-104

…have poor performance since the GeneXpert test has low sensitivity and the pleural fluid Ziehl-Neelsen staining and culture are time-consuming and commonly negative.

OR

…have limited diagnostic utility, as GeneXpert shows low sensitivity, and both staining and culture are time-consuming and frequently yield negative results.

106-108

…cut-off level of ≥ 35-40 U/L could be very helpful in diagnosing TPE with a high sensitivity, its specificity is low since high ADA levels, even ≥ 35-40 U/L are common in PPE, particularly the complicated, and in MPE, particularly the lymphomatous.

OR

…cut-off level of ≥35–40 U/L can be highly sensitive for diagnosing TPE, its specificity is limited, as elevated ADA levels are also common in PPE, particularly in complicated cases, and in MPE, especially lymphomatous types.

109-110

Diagnostic yield of closed pleural biopsy is not sufficiently high and its complications are not uncommon

OR

The diagnostic yield of closed pleural biopsy is not sufficiently high, and its complications are not uncommon.

112-113

…pH level is lower…

OR

…pH is typically lower…

113

…particularly the complicated…

OR

…particularly in complicated cases…

113-114

…but low level of pH could be present also...

OR

…however, low pH levels may also be observed...

114-116

Increased pleural fluid leukocytes with neutrophilic predominance is typical for PPE, but neutrophilic predominance could be not rarely present in MPE and in the acute phase of TPE.

OR

An increased pleural fluid leukocyte count with neutrophilic predominance is typical of PPE; however, neutrophilic predominance may also be observed, though less commonly, in malignant pleural effusion MPE and in the acute phase of TPE.

116-117

Moreover, pleural fluid glucose level less than 60 mg/dL could be present in PPE, in TPE and not rarely also in MPE.

OR

Moreover, pleural fluid glucose levels below 60 mg/dL may be observed in PPE, TPE, and, though less commonly, in MPE.

118-121

Many other markers have been suggested in the discrimination between various causes of exudative PE, such as tumor markers and cytokines, but the specificity and sensitivity of these parameters are not enough high for achieving a specific diagnosis of exudative PE.

OR

Many other markers, such as tumor markers and cytokines, have been proposed for distinguishing between various causes of exudative pleural effusion (PE); however, their sensitivity and specificity are not sufficiently high to enable a definitive diagnosis.

122-123

…pleural biopsy are regarded the golden tools for…

OR

…pleural biopsy are considered golden standard for…

123

…and can give excellent results…

OR

…and yield excellent results…

124

…particularly for MPE and TPE.

OR

…particularly in cases of MPE and TPE.

125

…for their invasive nature…

OR

…due to their invasive nature…

126

…and these procedures…

OR

…additionally these procedures…

131

…and in the starting of adequate…

OR

…and initiation of appropriate…

132

…usually brings to…

OR

…usually leads to…

141

…done with small groups of subjects, some of them…

OR

…conducted on small sample size; some of them…

145

…CRPr > 1 which it is impossible.

OR

…CRPr > 1, which is impossible.

148

…and has been demonstrated to be very useful in these differentiations.

OR

…where it has proven to be highly useful.

151-155

The target of the present study, which it is retrospective, performed on a relatively large group of subjects with exudative PE (282 subjects), much larger than most groups of previously published studies in the same regard, is to evaluate the efficacy rate of CRPs, CRPpf, CRPg, and CRPr in differentiation between various causes of exudative PE.

OR

The aim of this retrospective study, conducted on a relatively large cohort of 282 patients with exudative PE—a sample size larger than that of most previous studies—is to evaluate the diagnostic efficacy of serum CRPs, RPpf, CRPg, and CRPr in differentiating between various causes of exudative PE.

2.      Patients and Methods

2.1. Patients

158-166

The population of this study is composed of 282 subjects with exudative PE who were admitted to our Department of Internal Medicine, at Bnai Zion Medical Center between January 2000 and October 2016, have undergone PE drainage and extensive evaluation for exploring the etiology of their PE, and achieved definitive diagnosis of their PE etiology. Of these 282 subjects, 146 with age range of 24-92 years were with PPE, 126 with age range of 29-95 years were with MPE, and 10 subjects with age range of 23-86 years were with TPE. The PE was diagnosed PPE, MPE, or TPE in accordance with the worldwide accepted criteria presented in our recently published study.

OR

The study population consisted of 282 patients with exudative PE who were admitted to the Department of Internal Medicine at Bnai Zion Medical Center between January 2000 and October 2016. All patients underwent pleural fluid drainage and comprehensive diagnostic evaluation to determine the underlying etiology of their PE, resulting in a definitive diagnosis. Of these, 146 patients (aged 24–92 years) were diagnosed with parapneumonic PE (PPE), 126 (aged 29–95 years) with malignant PE (MPE), and 10 (aged 23–86 years) with tuberculous PE (TPE). The diagnoses of PPE, MPE, and TPE were established based on internationally accepted criteria, as described in our recently published study.

2.2.  Methods

168

Collection of relevant data was taken from the subjects’ files.

OR

Relevant data were extracted from the patients’ medical records.

169-170

…diagnosis of their PE as having PPE, MPE or TPE, and underwent measurement of levels of CRPpf and CRPs.

OR

…diagnosis of PE as PPE, MPE, or TPE, and who had undergone measurement of both CRPpf and CRPs levels.

171-172

Measurement of CRPpf and CRPs levels was done on a Cobas c 501 analyzer of Roche Diagnostics in accordance with the instructions of the Company.

OR

CRP levels were measured using a Cobas c 501 analyzer (Roche Diagnostics), following the manufacturer’s instructions.

2.3.   Statistical Analysis

179

…parameters are introduced as…

OR

…parameters are presented as…

180-181

To evaluate normality of the data, Kolmogorov-Smirnov test was done.

OR

The Kolmogorov–Smirnov test was used to assess the normality of data distribution.

181-182

We used unpaired Student’s t-test to compare between parametric groups.

OR

Unpaired Student’s t-tests were applied to compare parametric groups.

182-183

By Bonferroni correction test, p-values were corrected for multiple comparisons.

OR

Bonferroni correction was used to adjust p-values for multiple comparisons.

183-187

For detecting best cut-off 183 values (those with the maximal total accuracy, which was established by the SPSS software using the Youden index value which provides an optimal cut-off that balances between the sensitivity and the specificity values) for separations between groups, the receiver operating characteristic (ROC) analysis was used.

OR

Receiver operating characteristic (ROC) curve analysis was performed to determine optimal cut-off values for distinguishing between groups. These cut-off values, selected based on the highest total accuracy, were established using the Youden index, which balances sensitivity and specificity.

188-189

Calculation of sensitivity, specificity, negative predictive value (NPV), positive predictive value (PPV), total accuracy, area under the ROC curves, and odds ratio was done.

OR

Sensitivity, specificity, positive predictive value (PPV), negative predictive value (NPV), total accuracy, area under the ROC curve (AUC), and odds ratios were calculated.

3.      Results

193-198

The group of PPE contained 146 subjects, group of MPE contained 126 subjects, and group of TPE contained 10 subjects. Every subject in each group demonstrated CRPpf level lower than CRPs level. MPE group mean age was significantly higher than that of PPE group: 75.9 ± 10.2 years versus 69.6 ± 16.7 years, respectively (p < 0.003), and of the TPE group: 66.1 ± 19.2 years (p < 0.0009). Significant difference has not been demonstrated between PPE group and TPE group (p = 0.53).

OR

The PPE group included 146 subjects, the MPE group 126 subjects, and the TPE group 10 subjects. In all cases, pleural fluid CRP (CRPpf) levels were lower than serum CRP (CRPs) levels. The mean age of the MPE group was significantly higher than that of the PPE group (75.9 ± 10.2 years vs. 69.6 ± 16.7 years; p < 0.003) and the TPE group (66.1 ± 19.2 years; p < 0.0009). No significant age difference was observed between the PPE and TPE groups (p = 0.53).

4.      Discussion

442

…level but does not exceed it or even be equal to it.

Or

…level, but it never equals or exceeds it.

480

…differentiation between various causes…

OR

…differentiating between various causes…

480-483

This marker has been previously assessed only twice; for differentiation between transudative and exudative PE, and for differentiation between complicated and uncomplicated PPE, and proved to be very useful in these differentiations with a high efficacy rate.

OR

This marker has previously been evaluated in only two contexts: distinguishing between transudative and exudative PE, and between complicated and uncomplicated PPE. In both cases, it demonstrated high diagnostic utility.

484

...for differentiation between...

OR

...for differentiating between...

500-503

Our results stimulate performing further studies, prospective and retrospective, with large groups of subjects for investigating the role of this new marker in discrimination between various causes of exudative PE.

OR

Our findings support the need for further prospective and retrospective studies with larger patient cohorts to investigate the role of this marker in differentiating between various causes of exudative PE.

514-514

…and this is impossible.

OR

…which is impossible.

515

CRPpf level is always below that of CRPs…

OR

CRPpf level is consistently lower than the CRPs…

516-518

As stated above, CRP is synthesized mainly in the hepatocytes, its pleural fluid level depends mainly on its blood level, and cannot be equal to or exceed it.

OR

As stated above, CRP is synthesized primarily by hepatocytes, and its pleural fluid level depends largely on its serum concentration. Therefore, pleural CRP levels cannot equal or exceed serum levels.

540-543

The significant disadvantage of pleural fluid ADA is that its specificity is low where its level ≥ 35-40 U/L is also very common in PPE, particularly the complicated, and in MPE, particularly the lymphomatous.

OR

A major limitation of pleural fluid ADA is its low specificity, as levels ≥35–40 U/L are also commonly observed in PPE, particularly in complicated cases, and in MPE, especially those of lymphomatous origin.

5.      Conclusion

361-563

The present study results, demonstrated a high efficacy rate for CRPs and CRPpf in differentiation between various causes of exudative PE, but not for CRPr, which proved to be unhelpful in this differentiation.

OR

The results of the present study demonstrated high diagnostic value for CRPs and CRPpf in differentiating between various causes of exudative PE, whereas the CRPr was not useful for this purpose.

The TPE group (n=10) is too small for robust statistical inference. This limits generalizability and compromises power for comparisons involving TPE.

No adjustment for potential confounders like age, sex, comorbidities. This weakens the claim that CRP markers independently distinguish disease types.

The rejection of prior literature findings (e.g., CRPr >1) is valid, but should be addressed with more scientific depth regarding physiology, not just dismissive logic.

Numerous instances of awkward phrasing and minor grammatical errors. Comprehensive language editing is needed for clarity and professionalism.

Avoid redundancy (very helpful etc.)

Author Response

Reply to Reviewer 3 comment was uploaded in a PDF file.

Reviewer 4 Report

Comments and Suggestions for Authors

Dear author's 

your manuscript entitled Diagnostic Efficacy of C-Reactive Protein in Discrimination between Various Causes of Exudative Pleural Effusion: Disease Research Should not be Only for Rich is interesting  but is has numerous ethical concerns 1. In the present study, authors have used CRP as a biomarker to differentiate between various causes of exudative pleural effusion such as PPE, MPE and TPE.  My concern is that CRP is  a very nonspecific marker of inflammation and it is raised in all these conditions such as infective, inflammatory and malignant processes. CRP will not be able to differentiate between PPE, MPE and TPE.  2. In the present study, they have not mentioned how they have diagnosed PPE, MPE and TPE. for example PPE, by pleural aspiration and pleural fluid culture, MPE by pleural fluid cytology and fluid cell block or thoracoscopy or image guided pleural biopsy, and TPE by pleural fluid ADA level, pleural fluid MTB culture or Nuclic acid amplification tests on pleural fluid.  3. They have als mentioned with bold statement as patients effusion pleural fluid related invasive techniques for those having anxiety for invasive techniques can be diagnosed with CRP titer analysis. But, definitive diagnosis can be made only after documentation in specific tests such as pleural fluid analysis and histopathological analysis in PPE, MPE and TPE.  4. sample size is less to comment on major diagnostic roles in PPE, MPE and TPE.  5. add limitations to present study.  6. add inclusion and exclusion criteria in methodology. mention the confounding factors causing increase in CRP titer and add the paragraph mentioning that these conditions have been ruled out of these cases have been excluded.    revise the entire paper.    manuscript is of less clinical importance. revise and resubmit with mentioning of all necessary diagnostic tests used to diagnose because CRP is not diagnostic tests, neither sensitive nor specific for PPE, MPE and TPE. 

Author Response

Reply to Reviewer 4 comments was uploaded in a PDF file.

Round 2

Reviewer 4 Report

Comments and Suggestions for Authors

Dear authors 

you have made few corrections as per last review report, still; manuscript require more corrections and data additions 

  1. How your came to final diagnosis of various causes of exudative pleural effusion? As CRP is just a inflammatory markers without any diagnostic sensitivity and specificity
  2. which diagnostic modality were used to confirm parapneumonic effusion, Empyma, Tuberculous pleural effusion and malignant pleural effusion 
  3. you have also mentioned as, Pàtient’s not willing for thoracoscopy or plural biopsies can be diagnosed with CRP estimation 
  4. Overall merit if manuscript is very low, you have justified and mentioned as ADA is of lesser significance in diagnosis of tuberculous etiology!! I will suggest you to go through the published literature and see ADA is superior to CRP. Former one has definite sensitivity for Tuberculous pleural effusion and later one is neither sensitive nor specific

needs major revision 

Author Response

Point-by-Point response to the Reviewer's comments is present in an uploaded PDF file.

Round 3

Reviewer 4 Report

Comments and Suggestions for Authors

Dear authors

manus Is of very less clinical significance for current times in era of NAATS for TB, microbiome for Empyma and IHC for malignancy 

you have mentioned and recommendations CRP as diagnostic value which is neither sensitive nor specific 

Needs complete revision word to word of entire paper and remove line as it’s role in diagnosis of TPE, MPE, other etiologies